# A Novel Real-Valued DOA Algorithm Based on Eigenvalue

**DOI:** 10.3390/s20010040

**Published:** 2019-12-19

**Authors:** De-Sen Yang, Feng Chen, Shi-Qi Mo

**Affiliations:** 1Acoustic Science and Technology Laboratory, Harbin Engineering University, Harbin 150001, China; cfmail@hrbeu.edu.cn; 2Key Laboratory of Marine Information Acquisition and Security (Harbin Engineering University) Ministry of Industry and Information Technology, Harbin 150001, China; 3College of Underwater Acoustic Engineering, Harbin Engineering University, Harbin 150001, China; 17805014586@163.com

**Keywords:** DOA, super-resolution, real-valued processing, algorithm complexity, search range

## Abstract

To solve the high complexity of the subspace-based direction-of-arrival (DOA) estimation algorithm, a super-resolution DOA algorithm is built in this paper. However, in this method, matrix decomposition is required for each search angle. Therefore, in this paper, real-valued processing is used to reduce the scanning range by half, which is less effective in algorithm complexity. The super-resolution algorithm mainly uses the conservation of energy. By exploring the relationship between the covariance matrix and its complex conjugate, we constructed the real-valued matrix and introduced a real-valued searching source to make the operation of the matrix real-valued. Finally, the simulation experiments show that the proposed algorithm not only reduces the spectral search range by half but also has a higher angular resolution than the traditional algorithm.

## 1. Introduction

Direction-of-arrival (DOA) estimation is among the most important topics in array signal processing and shows potential applications in radar, sonar, and mobile communications [1,2,3,4,5]. The accuracy and resolution of DOA estimation are important indicators of the DOA estimation algorithm. The origin of this algorithm can be traced back to the conventional beam former (CBF) [6,7], which has a wide main lobe and an undulating side lobe. Given its inability to distinguish two adjacent source targets in the same main lobe, the resolution (i.e., the minimum discernible opening angle between two targets) is usually measured in terms of the smallest distinguishable angle. Angle resolution depends on wavelength and antenna aperture size) of the DOA estimation algorithm is subject to the Rayleigh limit. To improve the resolution of the DOA algorithm, a minimum variance distortionless response (MVDR) algorithm was developed [8,9,10]. The core idea of this algorithm is to maximize the gain in the target direction and minimize the output power of the array, which in turn will suppress the noise and power of the interfering signal. The MVDR algorithm also promotes the vigorous development of the DOA algorithm and provides the basis for developing super-resolution algorithms, such as the multiple signal classification (MUSIC) algorithm [11,12,13]. Although the MUSIC and Estimation of Signal Parameters via Rotational Invariance Techniques (ESPRIT) algorithms can achieve super-resolution, they are unable to meet the gradually increasing requirements for algorithm resolution and estimation accuracy in the field of engineering. To this end, many experts and scholars have intensively studied and proposed excellent algorithms with improved resolution or estimation accuracy. For instance, the maximum likelihood method proposed in [14] is now widely regarded as an optimal DOA estimator. With a sufficient SNR, the performance of this method can be approximated to the Cramér-Rao lower bound [15]. However, this method requires a multi-dimensional non-linear search for DOA information, thereby increasing its computation complexity and number of parameters. Meanwhile, Liu et al. [16] improved the root MUSIC algorithm based on coprime array, thereby greatly improving the resolution and estimation accuracy of this algorithm. In [17], the U-ESPRIT algorithm was built based on generalized least squares and demonstrated a better estimation performance than the traditional ESPRIT algorithm [18,19] under conditions with low SNR and snapshot coherent sources. However, this algorithm has strict requirements in array structure and lacks universality. In [20], Yan et al. proposed the real-valued MUSIC algorithm. This algorithm mainly utilizes the orthogonality of the real-valued noise subspace, thereby greatly reducing the amount of calculation. In [21], Zhou et al. studied the coprime array and applied it to the MUSIC algorithm, its performance has been greatly improved. Li et al. [22] studied the coprime array and proposed the U-root MUSIC algorithm, which has lower complexity and higher estimation accuracy than the conventional MUSIC and root MUSIC algorithms. With the rise of compressed sensing, some experts have proposed some algorithms based on compressed sensing [23]. However, despite their improved resolution and estimation accuracy, these algorithms mostly focus on eigenvectors and ignore eigenvalues. To address this gap [3], constructed a novel DOA algorithm based on subspace. But this algorithm requires complex matrix decomposition for each search angle. So, this paper constructs a novel DOA algorithm based on conservation of noise power and real-valued processing. The theoretical derivation proves that the computational complexity of the algorithm is effectively reduced by real-world operations and semi-spectral search.

The work done in this article can be summarized as follows. 1. This paper presents a concise expression for DOA estimation based on energy conservation and this algorithm can achieve super resolution. 2. In order to reduce the complexity of the algorithm, different from the existing algorithms, this paper selects the real-valued scanning steering vector and the real-valued covariance matrix, so that the scanning range is reduced by half. The rest of this paper is structured as follows. First, we constructed a simple spatial spectrum using energy conservation. Then the relationship between the covariance matrix and its complex conjugate is derived. By this association, the real-valued matrix is constructed and the search range is reduced by half, which greatly reduces the computational complexity.

## 2. Data Model

We assume *K* far-field narrow-band plane waves impinging on an uniform line array (ULA) comprising *M* array elements with an angle of arrival $\theta_{i},\theta_{i}=[\theta_{1},\theta_{2},\ldots ,\theta_{K}]$, where d denotes the spacing between each array element d=0.5γ and γ denotes wavelength. The direction of the incident plane wave is shown in Figure 1. The data received by the sensors can be expressed as:(1)X(t)=A(θ)S(t)+N(t),
where N(t) is the additive white Gaussian noise received by the sensors, S(t) is the *K*-by-1 signal vector matrix, X(t) is the entire array receiving dataset, and A(θ),θ∈[−90°, 90°] is the *M*-by-*K* steering vector matrix, which can be expressed as:(2)$$A(\theta )=[a(\theta_{1}),\ldots ,a(\theta_{K})].$$

By taking one of the angles $\theta_{k}$ and $\theta_{k}\in \theta_{i}$, the steering vector can be expressed as:(3)$$a(\theta_{k})={\left[1,e^{-j2\pi d\sin(\theta_{k})/\lambda },\ldots ,e^{-j2\pi (M-1)d\sin(\theta_{k})/\lambda }\right]}^{T}$$

Assume that the *K* far-field narrowband signals received by the array are independent of each other. In this case, the covariance matrix of the received data can be expressed as:(4)$$\begin{array}{ll} R & =E[X(t)X^{H}(t)]=A(\theta )R_{S}A^{H}(\theta )+\sigma_{n}^{2}I\\ \; & =\sum_{i=1}^{K}\sigma_{i}^{2}a(\theta_{i})a^{H}(\theta_{i})+\sigma_{{}^{n}}^{2}I \end{array}$$
where $\sigma_{n}^{2},\sigma_{i}^{2}$ denotes the noise and signal powers, respectively, H represents the conjugate transpose operation, $R_{S}=E[S(t)S^{H}(t)]$ represents the signal covariance matrix, and I represents the *M*-by-*M* unit matrix.

In addition to the above *K* targets [3], assume a far-field matching source with power $\sigma_{V}^{2}$ impinging on the array. The direction of the matching source is denoted by $\theta_{V}$. We build the matching covariance matrix ${\bar{R}}^{\theta_{V}}$ as follows:(5)$${\bar{R}}^{\theta_{V}}=R+\sigma_{V}^{2}a(\theta_{V})a^{H}(\theta_{V}),$$

So the spatial spectral function can be expressed as follows:(6)$$P_{S}(\theta_{v})=\frac{1}{\sum_{i=K+1}^{M}\left(\lambda_{i}-\bar{\lambda_{i}^{\theta_{v}}}\right)}.$$
where $\bar{\lambda_{i}}$ denotes the eigenvalues of ${\bar{R}}^{\theta_{V}}$.

## 3. Reduced-Complexity Algorithm

Eigendecomposition is applied on the covariance matrix R, and the eigenvalues are arranged in descending order. R can then be expressed as:(7)$$R=[U_{S}\;U_{N}]\left[\begin{array}{cc} \Sigma_{S} & \\ & \Sigma_{N} \end{array}\right]{\left[U_{S}\;U_{N}\right]}^{H},$$
where $U_{S}=[u_{1},\cdots ,u_{K}]$ includes the *K* column signal eigenvectors and is known as the signal subspace, whereas $U_{N}=[u_{K+1},\cdots ,u_{M}]$ includes the remaining M−K column noise eigenvectors and is known as the noise subspace. In Equation (7), $\Sigma_{S}$ and $\Sigma_{N}$ denote the diagonal matrices that comprise the signal and noise eigenvalues, respectively.

When the covariance matrix R is decomposed to obtain its eigenvalue, in theory, the eigenvalues should have the following characteristics:(8)$$\lambda_{1}\geq \lambda_{2}\geq \ldots \geq \lambda_{K}\geq \lambda_{K+1}=\lambda_{K+2}=\ldots =\lambda_{M},$$
(9)$$\lambda_{K+1}>\lambda_{K+2}>\ldots >\lambda_{M}.$$
This clearly shows that the eigenvalues corresponding to noise are divergent. To prevent these eigenvalues from diverging, the following covariance matrix is built:(10)$$R_{\mathrm{new}}=U\overset{⌢}{\Sigma }U^{H}.$$
where $\overset{⌢}{\Sigma }$ is a diagonal matrix that comprises eigenvalues that are constructed according to the following rules:(11)$$\left\{ \begin{array}{ll} {\overset{⌢}{\lambda }}_{i}=\lambda_{i} & ;\;(i=1,2,\ldots ,K)\\ {\overset{⌢}{\lambda }}_{i}=(\lambda_{K}+\lambda_{K+1})/2 & ;\;(i=K+1,K+2,\ldots ,M) \end{array}\right..$$
We build the matching covariance $R^{\theta_{V}}$ as follows:(12)$$R^{\theta_{V}}=R_{\mathrm{new}}+\sigma_{V}^{2}a(\theta_{V})a^{H}(\theta_{V}),$$
Which can be rewritten as:(13)$$R^{\theta_{V}}=\sum_{i=1}^{K}\sigma_{i}^{2}a(\theta_{i})a^{H}(\theta_{i})+\sigma_{V}^{2}a(\theta_{V})a^{H}(\theta_{V})+\sigma_{n}^{2}I.$$

The sources in the $\theta_{V}$ and $\theta_{i}$ directions are independent signals. $\theta_{V}$ can be divided into two cases, namely, $\theta_{V}\in \theta_{i}$ and $\theta_{V}\notin \theta_{i}$, where in both cases, the relationship between the eigenvalues of $R^{\theta_{V}}$ and $R_{\mathrm{new}}$ are examined.

**Theorem** **1:***If*$\theta_{V}\in \theta_{i}$, *the noise eigenvalues*$\lambda_{i}^{\theta_{V}}(i=K+1,\cdots ,M)$*of*$R^{\theta_{V}}$*are equal to the noise eigenvalues*${\overset{⌢}{\lambda }}_{i}(i=K+1,\cdots ,M)$*of*$R_{\mathrm{new}}$; *otherwise*, $\theta_{V}\notin \theta_{i}$, $\lambda_{i}{}^{\theta_{V}}(i=K+2,\cdots ,M)$, *and*${\overset{⌢}{\lambda }}_{i}(i=K+2,\cdots ,M)$*are all equal, whereas*$\lambda_{K+1}^{\theta_{V}}$*and*${\overset{⌢}{\lambda }}_{K+1}$*are not equal*.
$$\left\{ \begin{array}{l} \theta_{V}\in \theta_{i},\;\lambda_{K+1}^{\theta_{V}}={\overset{⌢}{\lambda }}_{K+1},\lambda_{K+2}^{\theta_{V}}={\overset{⌢}{\lambda }}_{K+2},\ldots ,\lambda_{M}^{\theta_{V}}={\overset{⌢}{\lambda }}_{M}\\ \theta_{V}\notin \theta_{i},\;\lambda_{K+1}^{\theta_{V}}\neq {\overset{⌢}{\lambda }}_{K+1},\lambda_{K+2}^{\theta_{V}}={\overset{⌢}{\lambda }}_{K+2},\ldots ,\lambda_{M}^{\theta_{V}}={\overset{⌢}{\lambda }}_{M} \end{array}\right.$$


*Proof*: See Appendix A.

Theorem 1 suggests that when $\theta_{V}\in \theta_{i}$, the noise eigenvalues $\lambda_{i}^{\theta_{V}}(i=K+1,\cdots ,M)$ are equal to the noise eigenvalues ${\overset{⌢}{\lambda }}_{i}(i=K+1,\cdots ,M)$. When $\theta_{V}\notin \theta_{i}$, the post M−K−1 noise eigenvalues $\lambda_{i}^{\theta_{V}}(i=K+2,\cdots ,M)$ are equal to the post M−K−1 noise eigenvalues ${\overset{⌢}{\lambda }}_{i}(i=K+2,\cdots ,M)$ and their (*K* + 1)-th eigenvalues are not equal. We use this property to build the following formula for DOA estimation:(14)$$P(\theta_{v})=\frac{1}{\lambda_{K+1}^{\theta_{V}}-{\overset{⌢}{\lambda }}_{K+1}}.$$

This formula is more physically meaningful than Equation (6) and effectively reduces the amount of calculation, but matrix decomposition is required for every search angle. Equation (14) mainly uses the eigenvalue ordering problem. When the scanning source and signal are consistent, the total number of signals does not increase, so the (*K* + 1)-th eigenvalue is the characteristic value of noise. On the contrary, when the scanning source and signal are not consistent, the total number of signals increases to *K* + 1, and the (*K* + 1)-th eigenvalue is the sum of the eigenvalue of the noise and the eigenvalue of the signal. To this end, we will mainly explain how to reduce its complexity.

### 3.1. Real-Valued Transform Algorithm

It is well known that the computational complexity of complex matrix decomposition is higher than the real matrix. Converting a complex matrix into a real matrix will undoubtedly reduce the complexity of the calculation. To do this, we carry out the following analysis.

We suppose K targets $s_{i}(i=1,2,\ldots ,K)$ with identical power symmetrical to the true position of the source and assume that the DOA is $-\theta_{i}(i=1,2,\ldots ,K)$. As shown in Equation (4), the covariance matrix can be seen as a function of angle information. Therefore, the matrix $R_{V}$ at the symmetry angle can be expressed as:(15)$$R_{V}=A(-\theta_{i})R_{s}A(-\theta_{i})+\sigma_{n}^{2}I.$$

Under the premise of meeting the requirements of the array $\forall \theta_{k}\in \theta_{i}$, the steering vector satisfies the following:(16)$$\begin{array}{l} a^{*}(\theta_{k})={\left[\exp(jd_{1}\sin(\theta_{k})),\cdots ,\exp(jd_{M}\sin(\theta_{k}))\right]}^{T}=\\ {\left[\exp(-jd_{1}\sin(-\theta_{k})),\cdots ,\exp(-jd_{M}\sin(-\theta_{k}))\right]}^{T}\\ \;=a(-\theta_{k}) \end{array}$$
where $d_{M}=(m-1)d$. By comparing Equations (4), (13) and (16), we have:(17)$$R_{V}=R^{*}.$$

Therefore, when the array steering vector satisfies the odd function, the covariance matrix is subjected to a complex conjugate operation, and the matrix obtained at this time can be regarded as a matrix containing symmetric angle information. Moreover, given that a complex conjugate operation is performed on $R_{\mathrm{new}}$, $R_{\mathrm{new}}^{*}$ also contains symmetric source information. Seeing that symmetric source information does not exist in practice, we label $R_{\mathrm{new}}^{*}$ as the pseudo data covariance matrix.

**Theorem** **2:***The covariance matrix*R*constructed by the sample data shares the same eigenvalue as that of the complex conjugated matrix*$R^{*}$*of*R, *and their subspaces show a complex conjugate relationship*.

*Proof*: See Appendix B.

A new matrix is constructed with the expression $R_{\mathrm{new}}+R_{\mathrm{new}}^{*}$. Therefore, through Theorem 2 $R_{\mathrm{new}}+R_{\mathrm{new}}^{*}$ can be further simplified as:
(18)Rnew+Rnew*=USΣ⌢SUSH+UNΣ⌢NUNH+US*Σ⌢S(US*)+UN*Σ⌢N(UN*)=USΣS^USH+US*ΣS^(US*)+2σ2I=2Re(Rnew)
where Re() represents the real part operation, $\hat{\Sigma_{S}}=\left[\begin{array}{ccc} \lambda_{1}-\sigma_{{}^{}}^{2} & & \\ & \ddots & \\ & & \lambda_{K}-\sigma_{{}^{}}^{2} \end{array}\right]$, $\sigma_{}^{2}={\overset{⌢}{\lambda }}_{i}=\frac{\left(\lambda_{K}+\lambda_{K+1}\right)}{2}$. Equation (18) shows that the constructed matrix 2Re(Rnew) increases the signal subspace by *K* columns, reduces the noise subspace by *K* columns, and doubles the noise eigenvalues. In this case, the number of signals reaches 2*K* at this time (including the real and mirror sources).

Even more interesting is that the matrix obtained in Equation (18) is a real matrix, which means that the complexity of the matrix operation will be reduced to one quarter of the complex matrix. However, in the Equation (12), $a(\theta_{V})$ is a complex vector, so the scan source is a complex matrix. If we still construct the matrix according to Equation (10), then the constructed matrix will be a complex matrix. Therefore, we must construct the scan source as a real matrix to ensure that the subsequent operations are real operations. We embody this in the operation of Equation (10):(19)Rnew=[Re(A)+jIm(A)]Rs⌢[Re(A)+jIm(A)]H+σ2I=[Re(A) Im(A)][Rs⌢00Rs⌢][ReH(A)ImH(A)]+σ2I+j[Re(A) Im(A)][Rs⌢00−Rs⌢][ImH(A)ReH(A)]
where Im() represents the imaginary part operation, $\overset{⌢}{R_{s}}$ is the power in Equation (10) and $\sigma^{2}$ is the noise power get from Equations (10) and (11). Re(Rnew) also can be rewritten as:(20)Re(Rnew)=[Re(A) Im(A)][Rs⌢00Rs⌢][ReH(A)ImH(A)]+σ2I.
So we can obtain a new steering vector:(21)Anew=[Re(A) Im(A)].

It is through this steering vector that we successfully construct the real target and the mirror target. So, we can replace the original scan vector with $A_{new}$. The advantage of this is that the scanned source is a real matrix, which does not involve complex calculations, effectively reducing complexity.

The matching matrix is constructed similar to Equation (12):(22)R˜θV=Re(Rnew)+σV2anew(θV)anewH(θV).
where anew=[Re(a) Im(a)]. We can deduce from Theorem 1 and Equation (18) that the (2*K* + 1)-th eigenvalue ${\tilde{\lambda }}_{2K+1}^{\theta_{V}}$ of the matching matrix ${\tilde{R}}^{\theta_{V}}$ demonstrates the following relationship with the (*K* + 1)-th eigenvalue ${\overset{⌢}{\lambda }}_{K+1}$ of $R_{\mathrm{new}}$:(23)$$\left\{ \begin{array}{l} {\tilde{\lambda }}_{2K+1}^{\theta_{V}}\approx {\overset{⌢}{\lambda }}_{K+1};\;\mathrm{when}\;\forall \theta_{V}\in -\theta_{i}\cup \theta_{i}\\ {\tilde{\lambda }}_{2K+1}^{\theta_{V}}\neq {\overset{⌢}{\lambda }}_{K+1};\;\mathrm{when}\;\forall \theta_{V}\notin -\theta_{i}\cup \theta_{i} \end{array}\right.$$

We then build the following spatial spectrum by using Equation (23):(24)$$f(\theta_{V})=\frac{1}{{\tilde{\lambda }}_{2K+1}^{\theta_{V}}-{\overset{⌢}{\lambda }}_{K+1}}$$

According to the spectral function constructed by using Equation (24), if the matching angle $\theta_{V}$ belongs to $\theta_{i}$ or $-\theta_{i}$, then the value of ${\tilde{\lambda }}_{2K+1}^{\theta_{V}}-{\overset{⌢}{\lambda }}_{K+1}$ approaches 0, and the spectral function forms a sharp peak at this angle. In the new spectral function, while retaining the target DOA information, a new artificially controllable peak is generated at its symmetrical position $\theta_{si}$ in order to display the symmetric source DOA information.

To obtain accurate DOA information, the DOA information obtained in the half spectrum must be discriminated to distinguish the symmetric source from the real source [24]. The following process is then applied on the angle obtained in the half spectrum:(25)$$\left\{ \begin{array}{l} f_{MUSIC}(\theta_{k})>>f_{MUSIC}(-\theta_{k}),\;\;\theta_{v}=\theta_{v}\cup \{\theta_{k}\}\\ f_{MUSIC}(\theta_{k})<<f_{MUSIC}(-\theta_{k}),\;\theta_{v}=\theta_{v}\cup \{-\theta_{k}\}\\ f_{MUSIC}(\theta_{k})\approx f_{MUSIC}(-\theta_{k}),\;\theta_{v}=\theta_{v}\cup \{\theta_{k},-\theta_{k}\} \end{array}\right..$$

By substituting the DOA information $\theta_{k}$ obtained in the half spectrum and the DOA information $-\theta_{k}$ at the symmetric position into the MUSIC algorithm function, this function comprises the noise subspace obtained via the eigendecomposition of R. We then compare the values of the two parameters of the MUSIC function given that the true angle will produce a spectral peak in the MUSIC algorithm and such a peak will not occur at the symmetric source angle. According to this property, we perform the discrimination by using Equation (25). When $f_{MUSIC}(\theta_{k})>>f_{MUSIC}(-\theta_{k})$, $f_{MUSIC}(\theta_{k})<<f_{MUSIC}(-\theta_{k})$ and $f_{MUSIC}(\theta_{k})\approx f_{MUSIC}(-\theta_{k})$, the true DOAs are $\theta_{k}$ and $-\theta_{k}$, respectively. Based on the above discrimination, the real and symmetric sources in the semi-spectrum can be distinguished from each other and a complete and accurate DOA information can be achieved.

### 3.2. Algorithm Pseudo Code

The pseudo code of the algorithm in this paper is as follows:

**Input**: Sensors signal X(t)

**Output**: DOA information $\theta =[\theta_{1},\theta_{2},\ldots ,\theta_{K}]$

    Initialization: ***R***
←
***X***(*t*)***X***^H^(*t*)/*n*

    Constructed a matrix $R_{\mathrm{new}}=U\overset{⌢}{\Sigma }U^{H}$, compute $U,\overset{⌢}{\Sigma }$ as Equation (11)

    Take the real part: $\mathrm{real}(R_{\mathrm{new}})=(R_{\mathrm{new}}+R_{\mathrm{new}}{}^{*})/2$


**Repeat**


    $\sigma_{V}^{2}=\mathrm{tr}(R_{\mathrm{new}})/M$

    Update ${\tilde{R}}^{\theta_{V}}=Re(R_{new})+\sigma_{V}^{2}a_{new}(\theta_{V})a_{new}{}^{H}(\theta_{V})$, compute $a_{new}(\theta_{V})$ as Equation (21)

    Update $f(\theta_{V})=\frac{1}{{\tilde{\lambda }}_{2K+1}^{\theta_{V}}-{\overset{⌢}{\lambda }}_{K+1}}$

**Until**$\theta_{V}$ is traversed

    $\left[\max,\theta ]=max(f(\theta_{V})\right)$

    By computing the MUSIC spectrum $f_{MUSIC}(\theta )$ as Equation (25), we can obtain the true DOA information.

### 3.3. Complexity Analysis

Table 1 compares the primary real-valued complexity between subspace-based method and the proposed algorithm. Since the matrix in subspace-based method is complex, its EVD requires $4JM^{3}$ flops, and the matrix in proposed method is real, its EVD requires $(J/2)M^{3}$ flops. Where J denotes searching points over [−90°, 90°], since the proposed method’s searching range is [−90°, 0°] or [0°, 90°], the searching points of the proposed method is (J/2). When calculating the spatial spectrum, the subspace-based method needs to perform M − K subtraction for each search point, but the method only needs to be used once(searching points of proposed method is (J/2)), so the calculation amounts of the two algorithms are J(M−K) and J/2 respectively. It can be obtained from Table 1. The proposed algorithm is one-eighth of the calculation amount of the subspace-based algorithm. We set J=180,M=12,K=2 for the two algorithms in a single DOA estimation experiment. Table 2 shows us the time required. It can be seen that the algorithm proposed in this paper has a great advantage in computing time.

## 4. Simulation Results

### 4.1. Performance Comparison

We assume that two incoherent sources of equal strength are incident on the array at 15° and 45° in the form of far-field plane waves. We select a uniform linear array of 12 sound pressure sensors arranged at half wavelength and the beam width at 3 dB is 11.56°. We compare the performance of the traditional MUSIC algorithm with that of the proposed algorithm when the SNR is 0 dB, $\sigma_{V}^{2}=tr(\hat{R})/M$ [25], and the number of snapshots is 100. The results are presented in Figure 2, and Figure 2a shows that the traditional MUSIC algorithm forms sharp peaks at 15° and 45°. Therefore, the DOA information of the target can be estimated accurately. Moreover, the proposed algorithm forms sharp peaks not only at 15° and 45° but also at its symmetry positions −15° and −45°, thereby verifying its correctness. In sum, the proposed algorithm and the traditional MUSIC algorithm are generally consistent in terms of side lobe height or main lobe width, thereby suggesting that these algorithms demonstrate the same performance under a large angular difference.

In order to verify the performance of the algorithm at different array element spacing, by setting d=0.25γ and d=γ, we can obtain Figure 2b. From Figure 2b, we can see that the resolution of the algorithm will become better with the increase of array element spacing (d=γ), but DOA ambiguity will appear. When the array element spacing becomes smaller (d=0.25γ), the algorithm resolution will become worse. Therefore, in order to obtain the best performance, the element spacing is generally selected to be equal to half the wavelength.

To further verify its performance, we test our proposed algorithm along with the MUSIC algorithm at small angular differences. Two uncorrelated narrowband sources are assumed to impinge the array from 15° and 20°, and the other experimental conditions are kept unchanged. We then compare the performance of these two algorithms under a small angular difference. The results are presented in Figure 3, which shows that the traditional MUSIC algorithm can only form one spectral peak in the spatial spectrum and is therefore unable to distinguish the two sources of 15° and 20°. In other words, the MUSIC algorithm fails under the experimental condition. Meanwhile, the proposed algorithm forms peaks at 15°, 20°, −15°, and −20°, thereby indicating that this algorithm forms peaks not only at the true DOA position but also at its symmetrical positions. Therefore, the correctness of the proposed algorithm is verified. This algorithm can also form spectral peaks under a small angular difference and can accurately distinguish the source, thereby highlighting its superiority over the traditional MUSIC algorithm. Such superiority can be mainly ascribed to the sensitivity of the eigenvalues to the changes in the DOA of the signal.

If the matching angle $\theta_{V}$ belongs to the target angle or the target symmetrical position angle, then the corresponding noise eigenvalue of ${\tilde{\lambda }}_{2K+1}$ is equal to ${\overset{⌢}{\lambda }}_{K+1}$, and the matching angle $\theta_{V}$ does not belong to the target angle or target symmetrical position angle. In this case, ${\tilde{\lambda }}_{2K+1}$ immediately becomes a signal-corresponding eigenvalue that is much larger than ${\overset{⌢}{\lambda }}_{K+1}$.

### 4.2. Statistical Properties Comparison

Given that the above experiments are all single experiments, their results cannot effectively reflect the performance of the proposed algorithm. Therefore, while keeping the aforementioned experimental conditions unchanged, we select two non-coherent sources with an angular difference of $\Delta \theta =5^{{}^{\circ}}$ for 100 Monte Carlo experiments. We use the proposed algorithm, the MUSIC symmetrical compressed spectrum (MSCS), the MUSIC algorithm, and NSP for the simulation experiments. We set the SNR from −10 dB to 2 dB each time until reaching 10 dB and then compute the root mean squared error (RMSE) and success rate of the four algorithms for each SNR. The RMSE statistics of the three algorithms under different SNRs are shown in Figure 4, from which we can observe that when the SNR is lower than −8 dB, the RMSE of the four algorithms are roughly the same, thereby suggesting that the four algorithms show a poor estimation performance when the SNR is too low. However, when the SNR increases, the RMSE of the proposed algorithm, NSP, MSCS, and MUSIC algorithms is rapidly reduced. In sum, the RMSE of the proposed algorithm is less than that of the NSP, MSCS, and MUSIC algorithms, thereby suggesting that the proposed algorithm outperforms the other algorithms under this experimental condition.

We now assume that the deviation between the estimated and true angles is less than 1°. Under this condition, the proposed algorithm is considered successful. Therefore, the resolution success rate can be defined as the ratio of the estimated success times to the number of Monte Carlo trials. Figure 5 shows that the success rate of the three algorithms increases along with SNR. When the SNR is sufficiently high, the success rates of the three algorithms tend to be consistent. Overall, the success rate of the proposed algorithm is higher than that of the traditional MUSIC, MSCS, and NSP algorithms.

We keep the aforementioned experimental conditions unchanged and set the number of sensors from 6 to 12 for 100 Monte Carlo experiments. Figure 6 shows that the success rate of the three algorithms increases along with the number of sensors. Overall, the algorithm of this paper has the best performance among the four algorithms.

We now evaluate the performance of the proposed algorithm with that of the NSP, MSCS, and MUSIC algorithms under varying angular differences. We use a DOA set of 15° and 20° to generate an angular difference of $\Delta \theta =5^{{}^{\circ}}$, and then use another DOA set of 15° and 35° to generate an angular difference of $\Delta \theta =20^{{}^{\circ}}$. A total of 100 Monte Carlo experiments are then performed. The number of snapshots in the experiments is initially set to 10 and increases by 30 each time until reaching 310. The results are shown in Figure 7, which reveals that at an angular difference of $\Delta \theta =20^{{}^{\circ}}$, the four algorithms have consistent success rates while approaching one another at different numbers of snapshots. Meanwhile, when the angular difference is $\Delta \theta =5^{{}^{\circ}}$, the success rate of the proposed algorithm is higher than that of the traditional MUSIC, NSP, and MSCS algorithms under different numbers of snapshots. By comparing the performance of these algorithms under angular differences of $\Delta \theta =5^{{}^{\circ}}$ and $\Delta \theta =20^{{}^{\circ}}$, we find that the proposed algorithm achieves a high angular resolution and is most suitable for estimating DOA under a small angular difference.

We then set the SNR to 10 dB, the number of snapshots to 100, one of the target angles to 15°, and the other target angles to 17.5°, 18°, 18.5°, 19°, 19.5°, and 20° before conducting 100 Monte Carlo simulation experiments. The results are shown in Figure 8, which reveals that the proposed algorithm is superior to the other algorithms under a small angular difference. In sum, the proposed algorithm shows a better resolution compared with the other algorithms.

### 4.3. Actual Data Processing

For the actual data processing, we use data collected from the Songhua Lake experiment and an 8-element line array with an element spacing of 0.0952 m, 1 target, 30 kHz signal frequency, and 79 snapshots. The power amplifier is BK2713, the signal source is Aglient33522A, the collector is NI cRIO-9031, and the hydrophone is 8106. After the received “wav” file is read by the corresponding reading program, the subsequent processing is performed. The proposed algorithm is employed for the processing, and the results are presented in Figure 9, which shows that two bright points are formed and that the signal lasts for approximately 5 seconds. These results are consistent with our launch duration, thereby indicating the effectiveness of the proposed algorithm.

## 5. Conclusions

In this paper, the special relationship between the eigenvalues of the matching covariance matrix and those of the covariance matrix are derived to construct a new spatial spectrum for DOA estimation. We then perform a pseudo-data reconstruction to reduce the computational complexity of the proposed algorithm. The simulation results demonstrate that the proposed algorithm is superior to the MUSIC and NSP algorithms in terms of resolution and greatly contributes to distinguishing those targets that are close to one another in space. However, the proposed algorithm needs to perform eigendecomposition in each search. The MUSIC and NSP algorithms do not strictly require array steering vectors, whereas the proposed algorithm requires the delay function of the steering vector to be an odd function. The proposed algorithm can also be applied if the steering vectors of some arrays satisfy the odd function after a spatial transformation.

## 6. Patents

The authors would like to thank the editor and reviewers for reading and comments. This work was supported by the Program for Chang Jiang Scholars and Innovative Research Team in University of Ministry of Education of China (Grant No.IRT_16R17) and Submarine autonomous weapons platform in the deep sea (Grant No.KY10500180037).

## Figures and Tables

**Figure 1 sensors-20-00040-f001:**
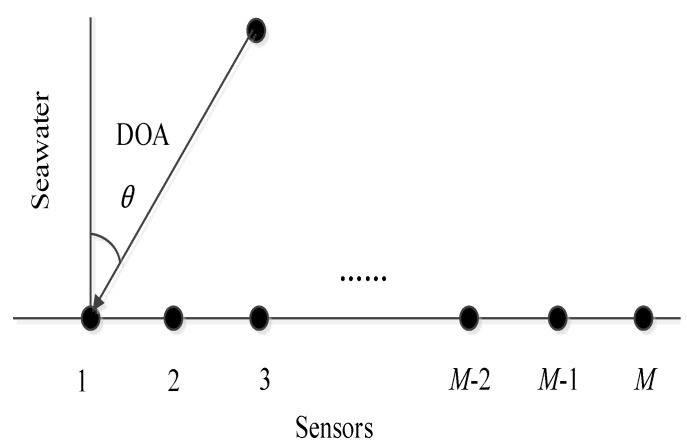
Array receiving model.

**Figure 2 sensors-20-00040-f002:**
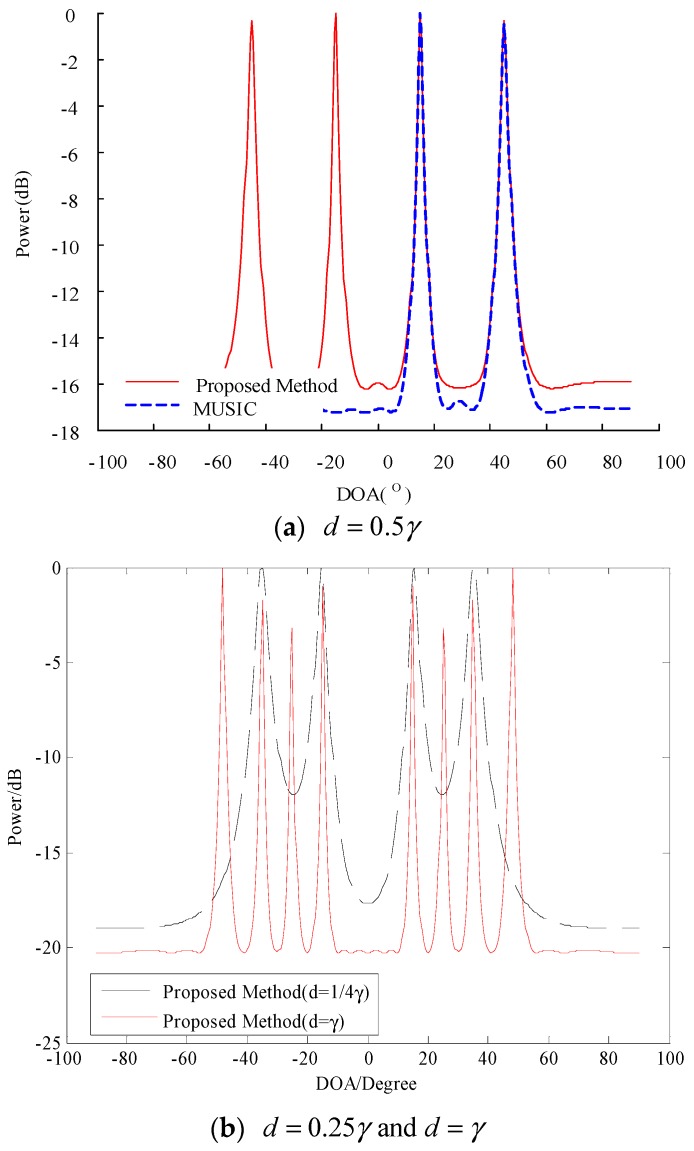
Spatial spectrum of the two algorithms under directions-of-arrival (DOAs) of 15° and 45°.

**Figure 3 sensors-20-00040-f003:**
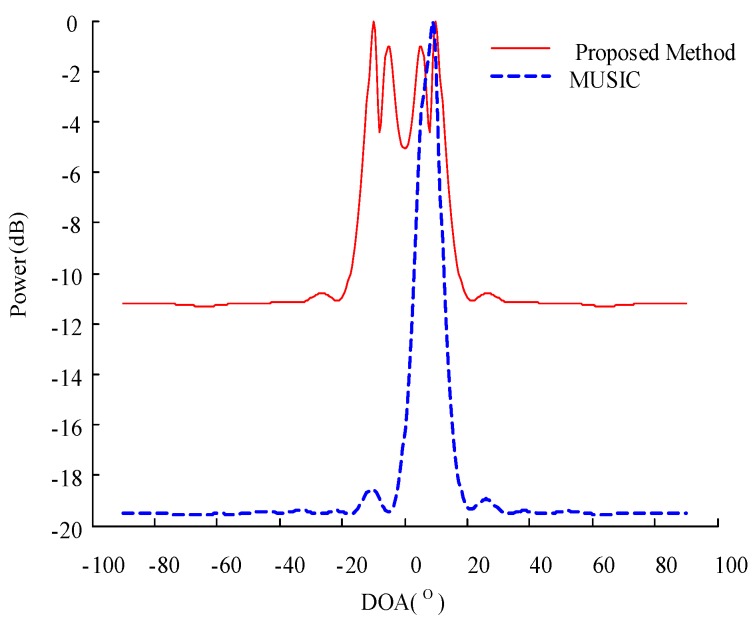
Comparison of the spatial spectra of different algorithms under a small angular difference.

**Figure 4 sensors-20-00040-f004:**
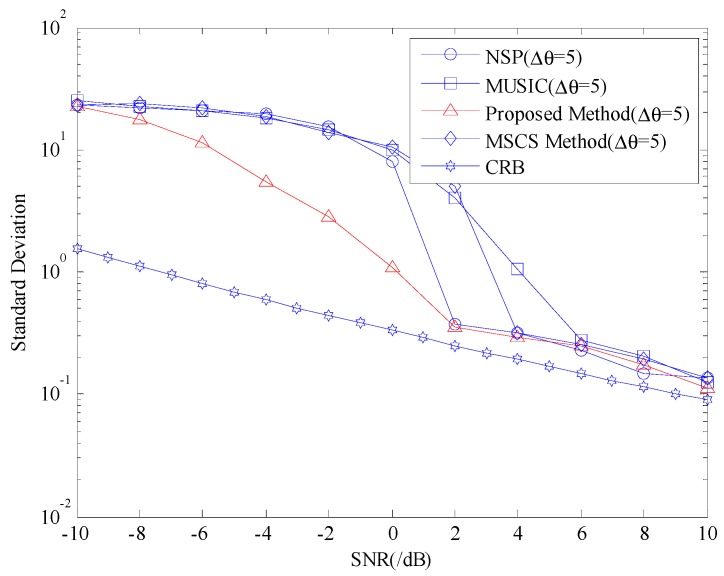
Comparison of the root mean squared errors (RMSEs) of the four algorithms under different signal-to-noise ratios (SNRs).

**Figure 5 sensors-20-00040-f005:**
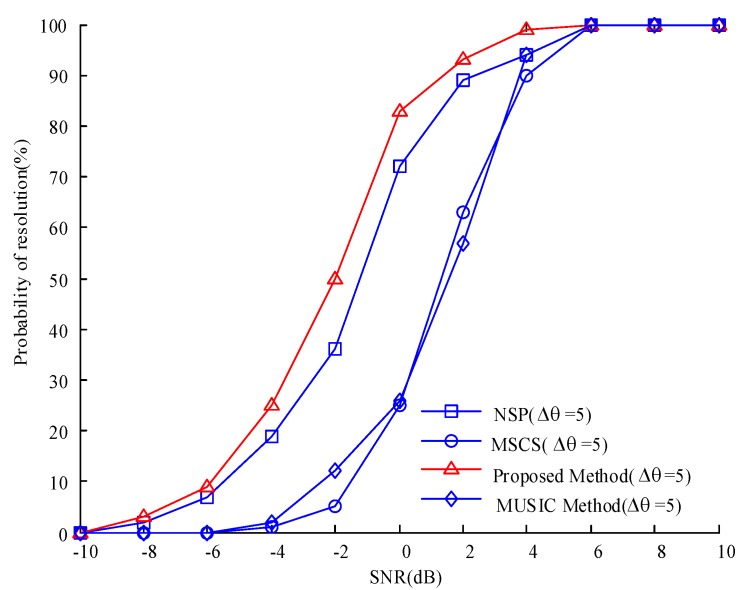
Comparison of the resolution success rates of the four algorithms under different SNRs.

**Figure 6 sensors-20-00040-f006:**
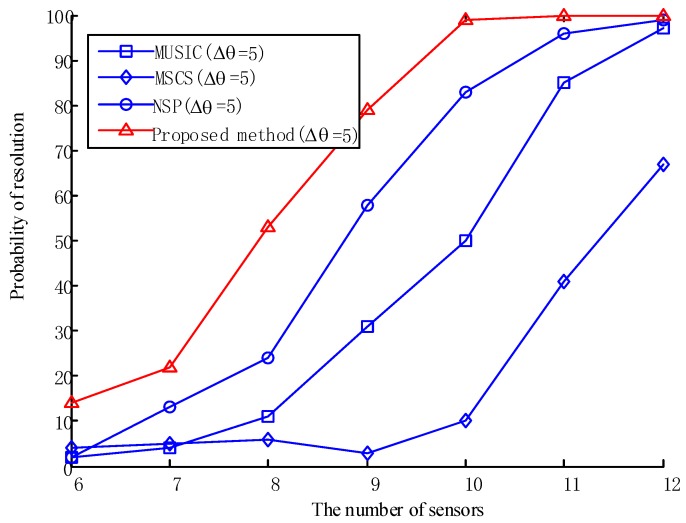
Comparison of the resolution success rates of the four algorithms with a different number of sensors.

**Figure 7 sensors-20-00040-f007:**
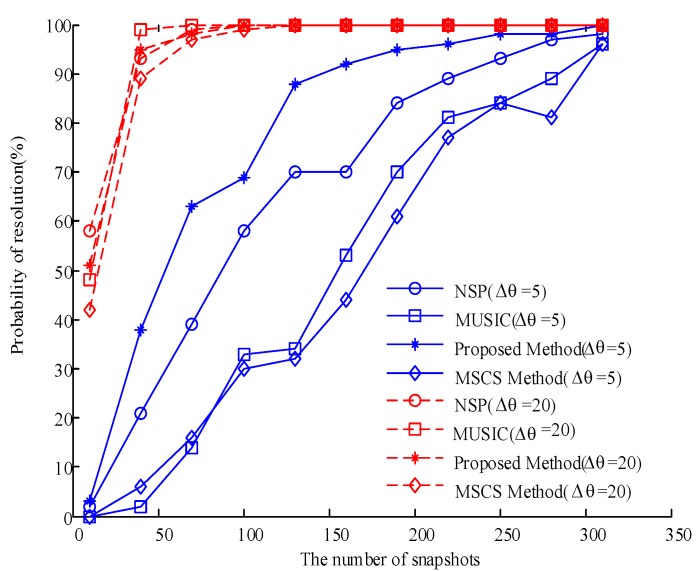
Comparison of the resolution success rates of the four algorithms under different snapshot numbers and angular differences.

**Figure 8 sensors-20-00040-f008:**
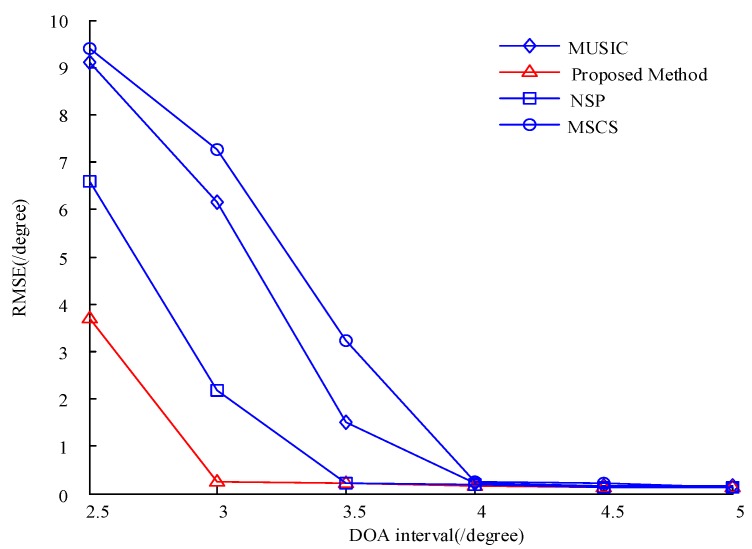
Comparison of the RMSE of the four algorithms under different DOA intervals.

**Figure 9 sensors-20-00040-f009:**
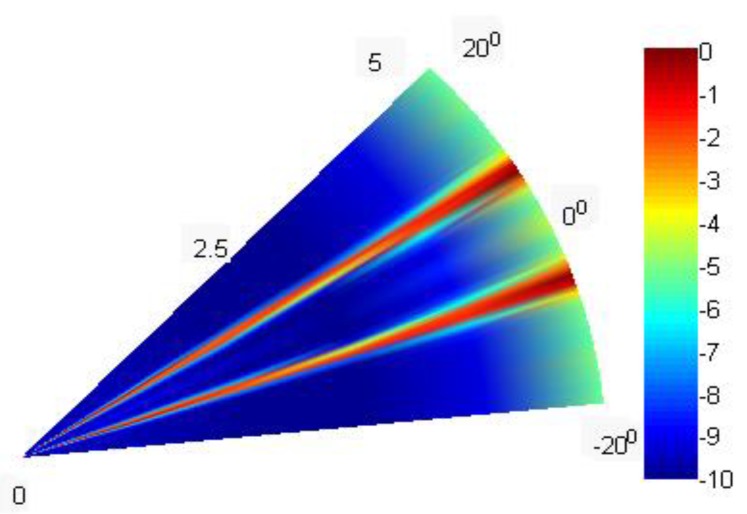
Bearing-time records of the proposed algorithm.

**Table 1 sensors-20-00040-t001:** Comparison of computational complexity.

Algorithms	Computations
subspace-based method	$4\times [JM^{3}]+J(M-K)$
Proposed method	$[JM^{3}+J]/2$

**Table 2 sensors-20-00040-t002:** Comparison of calculating time.

Algorithms	Time (Second)
subspace-based method	0.0513
Proposed method	0.0076

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
