# Peer review of "A Novel Real-Valued DOA Algorithm Based on Eigenvalue"

_sensors, 2019, doi:10.3390/s20010040_

Round 1

Reviewer 1 Report

The authors present a new algorithm for Direction-of-Arrival estimation. The proposed technique is an ameliorated version of the MUSIC algorithm. After the description of it also simulation results are presented.

However some questions and/or remarks remain:

General remark: please clarify the difference between lambda as wavelength and lambda as eigenvalue

L14: which is less effective in algorithm complexity: you mean that the real-valued processing is less effective or the reduction of the scanning range?

L64-65: please rephrase sentence, it contains 3 times the d

L93: please explain how you can make a difference between noise subspace and signal subspace

L101-102: please prove that the noise eigenvalues are always divergent

L109: can you explain practically what happens in that formula?

L120: can you explain practically what happens in that formula?

L128: more physically: please lengthen the explanation

L227: give also some simulation results where the distance is not half wavelength

L236: what do you mean with the sentence: thereby verifying its correctness?

L240: the proposed method does not make distinction between an arrival angle of -20° and 20°. How to make this distinction when needed in practical applications?

L259: the proposed method stays most of the time around -11 dB. When can this level be higher? And what happens then?

L309: can you give some calculated & simulated values (in seconds) for the complexity of the algorithm?

Some typing errors:

L28: discernable -> discernible

L58-59: sentence misses verb

L135: sentence must end with a final point

L144: Where -> where

L313: 8106.After -> 8106. After

Author Response

Reviewer 1

Thank you for your comments, I will reply to them below:

The authors present a new algorithm for Direction-of-Arrival estimation. The proposed technique is an ameliorated version of the MUSIC algorithm. After the description of it also simulation results are presented.

However some questions and/or remarks remain:

General remark: please clarify the difference between lambda as wavelength and lambda as eigenvalue

Response: Thank for your comments, I have replaced the wavelength with other letters.

L14: which is less effective in algorithm complexity: you mean that the real-valued processing is less effective or the reduction of the scanning range?

Response: Thank for your comments. L14 should be changed to be “more effective”. First, the real valued operation is less complex than the complex calculation. Second, the algorithm in this paper halves the search range. Therefore, the algorithm in this paper is more advantageous in terms of computation.

L64-65: please rephrase sentence, it contains 3 times the d

Response: Thank for your comments. I have modified it in the article.

L93: please explain how you can make a difference between noise subspace and signal subspace

Response: Thank for your comments. First, we sort the eigenvalues and arrange the eigenvectors in the order of the eigenvalues. When we know the number of sources K, then the eigenvectors corresponding to the K large eigenvalues will form the signal subspace, and the remaining eigenvectors are the noise subspaces.

L101-102: please prove that the noise eigenvalues are always divergent

Response: Thank for your comments. I can't prove that noise always diverges, but a lot of literature gives this conclusion, such as "Source Enumeration Via MDL Criterion Based on Linear Shrinkage Estimation of Noise Subspace Covariance Matrix. IEEE TRANSACTIONS ON SIGNAL PROCESSING, VOL. 61, NO. 19, OCTOBER 1, 2013 " Eqs 3 and 5 give the same conclusion as L101-102.

L109: can you explain practically what happens in that formula?

Response: Thank for your comments. First we construct the  matrix according to Eq. 10, is just a scanning angle,  is the value we give. When scanning, you only need to construct the matrix according to Eq. 12, and then decompose the matrix to obtain the required eigenvalues, and then calculate according to Eq.14. In practice, no additional source is required, so information collection is the same as conventional collection methods.

L120: can you explain practically what happens in that formula?

Response: Thank for your comments. This can be explain by Eq. 12. When the scan angle and the target angle are the same, we can consider that the scan source and the target are coincident, and the scaning source’s energy added the target’s energy, but the number of signals does not increase, and there are still K signals. Therefore, the following M-K eigenvalues are eigenvalues of noise. On the contrary, one more signal is added, and the number of signals becomes K+1, so the K+1th eigenvalue is the noise power plus the signal power, which is no longer equal to the noise power. But in both cases, the remaining eigenvalues are noisy and therefore equal. Below I give some formulas to describe the changes in eigenvalues.

Where  is the energy of scanning source.

L128: more physically: please lengthen the explanation

Response: Thank for your comments. I have modified it in the article.(L138)

L227: give also some simulation results where the distance is not half wavelength

Response: Thank for your comments. I have modified it in the article. (Fig. 2b)

L236: what do you mean with the sentence: thereby verifying its correctness?

Response: Thank for your comments. Because the purpose of this algorithm is to halve the search range, we can make subsequent judgments only if symmetrical peaks are formed in the half-spectrum range. So when the real angle is 15 and 35, we hope that -15 and -35 also form spectral peaks, so No matter when we scan [0 90] or the scan is [-90 0], we can get the spectral peaks. After getting the spectral peaks, we can get the true signal angle by Eq. (25).

L240: the proposed method does not make distinction between an arrival angle of -20° and 20°. How to make this distinction when needed in practical applications?

Response: Thank for your comments. In fact, when we know the number of sources, this is no longer a problem. For example, we only find -20 peaks in [-90 0], and when we know the number of sources is 2, which shows that there is another target symmetrical position. We can also use Eq. 25 to explained. When we find a spectrum peak at -20 and bring it into Eq.25, it satisfies Case 3, so we can know that 20 is also the target. So the decision of -20 and 20 belongs to the third case of Eq. 25.

L259: the proposed method stays most of the time around -11 dB. When can this level be higher? And what happens then?

Response: Thank for your comments. -10dB is a simulation of SNR. For example, at -15dB, the effect we get will be worse than -10dB, and it is possible that all four methods fail. Of course, if you want -10dB to get better results, we can increase the number of array elements or the number of snapshots. This experiment was performed to illustrate that with the increase of the SNR, the algorithm in this paper is superior to other algorithms.

L309: can you give some calculated & simulated values (in seconds) for the complexity of the algorithm?

Response: Thank for your comments. I have modified it in the article. (Table 2)

Some typing errors:

L28: discernable -> discernible

L58-59: sentence misses verb

L135: sentence must end with a final point

L144: Where -> where

L313: 8106.After -> 8106. After

Response: Thank for your comments. I have modified it in the article.

Reviewer 2 Report

How does the algorithm distinguish whether the angles of arrival are negative or positive?, since, when obtaining a real matrix from the complex, the angles and their symmetrical appear

By keeping the real part, what information is being lost?

You must specify the beamwidth at 3 dB of the array being used, for experiments via simulation and for real experiments

The new algorithm must be validated according to the number of sensors in the array and not only according to the SNR level.

Author Response

Reviewer 2

How does the algorithm distinguish whether the angles of arrival are negative or positive?, since, when obtaining a real matrix from the complex, the angles and their symmetrical appear

Response: Thank for your comments. In fact, when we know the number of sources, this is no longer a problem. For example, there are two signals here, their DOAs are 20 and -20. When we only find a peak at-20 in [-90 0], and due to the number of sources is 2, which shows that there is another target symmetrical position. We can also use Eq. 25 to explain. When we find a spectrum peak at -20 and bring it into Eq.25, since 20 is also a true angle, it satisfies Case 3, so we can know that 20 is also the target. So for symmetric targets, we can distinguish them by the third case of Eq. 25.

By keeping the real part, what information is being lost?

Response: Thank you for your comments. From Eq. 20 we can see that there is no loss in terms of information. Eq. 20 shows that the new steering vector contains the real and imaginary parts of the original steering vector, so the performance of the algorithm will not reduce. The original steering vector can be regarded as the projection of the steering vector on the x and y axes, that is, the real and imaginary parts, and the real and imaginary parts determine the size and direction of the steering vector. Therefore, when the target is incident at 200, the algorithm will also form a spectral peak at -20 0, which is also the core of this algorithm.

You must specify the beamwidth at 3 dB of the array being used, for experiments via simulation and for real experiments

Response: Thank for your comments. I have modified it in the article. (L249). From the figure we see that the beam width is 11 degrees, but the algorithm can distinguish targets within 5 degrees. I think the algorithm in this article is not a beam algorithm, so the resolution of the algorithm must be less than the beam width at 3db.

The new algorithm must be validated according to the number of sensors in the array and not only according to the SNR level.

Response: Thank for your comments. I have modified it in the article. (Fig 6)

Reviewer 3 Report

In this paper, a real-valued super-resolution direction of arrival (DOA) estimation algorithm is proposed by investigating the special relationship between the eigenvalues. The theoretical analysis and proofs are provided with the simulation results carried out with both numerical simulation and actual data processing. I have the following comments to improve this paper.

1. The main contribution of this paper should clarify in the Introduction. In current version, only the steps of the proposed algorithm was provided. Moreover, the substantial difference with the state-of-art real-valued DOA estimation algorithms should be claimed.

2. The literature review can be updated, including the recent advances on real-valued DOA estimation, super-resolution DOA estimation, which pursuit the low complexity as this paper did, such as,

[R1] Real-valued MUSIC for efficient direction estimation with arbitrary array geometries, IEEE Transactions on Signal Processing, 2014.

[R2] Direction-of-arrival estimation for coprime array via virtual array interpolation, IEEE Transactions on Signal Processing, 2018.

[R3] Two sparse-based methods for off-grid direction-of-arrival estimation, Signal Processing, 2018.

3. The title seems not appropriate. First, the proposed algorithm does not belong to a generalized framework for a general class of super-resolution DOA estimation with real-valued processing. Second, the state-of-art algorithm such as [R1] can also categorized to real-valued super-resolution DOA estimation. Therefore, a more specific title is preferred.

4. In (6), the right-side is not a function of the direction theta, please clarify. So does the equation (14).

5. In (11), why the noise eigenvalue are set as the mean of k-th and (k+1)-th eigenvalue? Usually, the noise eigenvalue is much smaller that the signal eigenvalue. Moreover, what is its relationship with the conclusions in Theorem 1, which relates to these eigenvalues?

6. Usually, the EVD take an order of 3 computation plops. I would suggest the authors to double check the conclusions in 3.3.

7. Usually, the MUSIC algorithm infinitely approaches the CRB. Please double check the CRB in Fig. 4. Maybe another square is required on the current CRB, since the Y-axis represents the RMSE.

8. Several grammar typos. A careful revision is required. Moreover, the vectors and matrices are usually in boldface. Cramer-Rao --> Cram\'{e}r-Rao.

Author Response

Reviewer 3

In this paper, a real-valued super-resolution direction of arrival (DOA) estimation algorithm is proposed by investigating the special relationship between the eigenvalues. The theoretical analysis and proofs are provided with the simulation results carried out with both numerical simulation and actual data processing. I have the following comments to improve this paper.

The main contribution of this paper should clarify in the Introduction. In current version, only the steps of the proposed algorithm was provided. Moreover, the substantial difference with the state-of-art real-valued DOA estimation algorithms should be claimed.

Response: Thank for your comments. I have modified it in the Introduction.

The literature review can be updated, including the recent advances on real-valued DOA estimation, super-resolution DOA estimation, which pursuit the low complexity as this paper did, such as,

 [R1] Real-valued MUSIC for efficient direction estimation with arbitrary array geometries, IEEE Transactions on Signal Processing, 2014.

[R2] Direction-of-arrival estimation for coprime array via virtual array interpolation, IEEE Transactions on Signal Processing, 2018.

[R3] Two sparse-based methods for off-grid direction-of-arrival estimation, Signal Processing, 2018.

Response: Thank for your comments. I have modified it in the Introduction.

The title seems not appropriate. First, the proposed algorithm does not belong to a generalized framework for a general class of super-resolution DOA estimation with real-valued processing. Second, the state-of-art algorithm such as [R1] can also categorized to real-valued super-resolution DOA estimation. Therefore, a more specific title is preferred.

Response: Thank you for your comments. I have modified it in the title.

In (6), the right-side is not a function of the direction theta, please clarify. So does the equation (14).

Response: Thank for your comments. I have modified Eqs.6 and 14

In (11), why the noise eigenvalue are set as the mean of k-th and (k+1)-th eigenvalue? Usually, the noise eigenvalue is much smaller that the signal eigenvalue. Moreover, what is its relationship with the conclusions in Theorem 1, which relates to these eigenvalues?

Response: Thank for your comments. Before I start answering, I gave a few formulas. I hope that through these formulas, it will be more convenient for my interpretation.

We can see from the above formula that the performance of the algorithm does not directly depend on K-th and (K+1)-th eigenvalues. I add them for two purposes. 1. I want the eigenvalues of the noise not to diverge. 2 I add the signal eigenvalue to the noise eigenvalue, which can make the algorithm more robust, because the noise eigenvalue has a greater impact on the number of snapshots and SNR. Adding signal eigenvalues can make the eigenvalues of  not change drastically

Usually, the EVD take an order of 3 computation plops. I would suggest the authors to double check the conclusions in 3.3.

Response: Thank for your comments. I have modified in 3.3 (table I). If we use the FSD [R1] method, the EVD calculation amount is . To make the article easier to understand, I have modified it as required. This will not affect the final conclusion.

[R1]B. D. Rao and K. V. S. Hari, “Performance analysis of root-MUSIC,” IEEE Trans. Acoust., Speech, Signal Process., vol. 37, no. 12, pp. 1939–1949, Dec. 1989.

Usually, the MUSIC algorithm infinitely approaches the CRB. Please double check the CRB in Fig. 4. Maybe another square is required on the current CRB, since the Y-axis represents the RMSE.

Response: Thank for your comments. Because I started to set the scan step size, the error caused was larger. To reduce the error, I have reduced the search step size. The result is shown in Fig. 4.

Several grammar typos. A careful revision is required. Moreover, the vectors and matrices are usually in boldface. Cramer-Rao --> Cram\'{e}r-Rao.

Response: Thank for your comments. I have modified in the paper.

Round 2

Reviewer 1 Report

No further comments

Author Response

Dear Professor

   Thank for your comments

Reviewer 2 Report

Changes has been done!!

Author Response

Dear Professor

   Thank for your comments

Reviewer 3 Report

The reviewer appreciates the authors' revision. There is only one thing that the authors need to confirm before publication version. In particular, the title of this paper is not complete or correct in current version. Please pay attention. 

No other comments.

Author Response

Dear Professor

I have changed the title of the article to "A novel real-valued DOA algorithm based on eigenvalue".